# The Reporting, Use, and Validity of Patient-Reported Outcomes in Multiple Myeloma in Clinical Trials: A Systematic Literature Review

**DOI:** 10.3390/cancers14236007

**Published:** 2022-12-06

**Authors:** Sam Salek, Tatiana Ionova, Esther Natalie Oliva, Marike Andreas, Nicole Skoetz, Nina Kreuzberger, Edward Laane

**Affiliations:** 1School of Life and Medical Sciences, University of Hertfordshire, Hatfield AL10 9AB, UK; 2Quality of Life Monitoring Department, Saint-Petersburg State University Hospital, 190103 Saint-Petersburg, Russia; 3Grande Ospedale Metropolitano Bianchi Melacrino Morelli, 89124 Reggio Calabria, Italy; 4Evidence-Based Medicine, Department I of Internal Medicine, Faculty of Medicine and University Hospital Cologne, University of Cologne, 50937 Cologne, Germany; 5Hematology-Oncology Clinic, Tartu University, 50406 Tartu, Estonia; 6Kuressaare Hospital, 93815 Kuressaare, Estonia

**Keywords:** multiple myeloma, patient-reported outcomes, quality of life, symptoms, systematic review, clinical trials, validation studies

## Abstract

**Simple Summary:**

In multiple myeloma, a type of blood cancer, the measurement of the outcomes of treatment reported by patients (patient-reported outcome measurement) is useful to reveal how treatment affects the quality of life, symptoms, and side effects of treatment to evaluate the benefit-risk balance of a particular drug or drug combination. We evaluated the research reported in the literature to identify and evaluate patient-reported outcome measures used in clinical trials. The most frequently used instrument was the generic questionnaire, named EORTC QLQ-30, used for cancer patients. The second most frequently used tools were the multiple myeloma-specific EORTC-MY20 questionnaires and the generic EQ-5D. Only 19 instruments reported in the literature were consistent with the trial design. The findings indicate that the measurement of patient-reported outcomes in clinical trials for multiple myeloma patients is underutilised, underreported, and often inconsistent. Thus, guidelines for the appropriate use and reporting of such questionnaires are needed to ensure standardisation in the selection and reporting of patients’ views. This report is a valuable reference for the appropriate use of questionnaires for professionals performing clinical trials to ensure accurate and appropriate reporting of results for multiple myeloma patients and their caregivers and patient advocates.

**Abstract:**

Background: Patient-reported outcomes (PROs) are becoming increasingly important in supporting clinical outcomes in clinical trials. In multiple myeloma (MM), PRO measurement is useful to reveal how treatment affects physical, psychosocial, and functional behaviour as well as symptoms and treatment-related adverse events to evaluate the benefit-risk ratio of a particular drug or drug combination. We report the types of PRO instruments used in MM, the frequency in which they are utilised in randomised controlled trials (RCTs), and the consistency of their reporting. Methods: The European Hematology Association (EHA) supports the development of guidelines for the use of PROs in adult patients with haematological malignancies. The first step is the present systematic review of the literature. MEDLINE and CENTRAL were searched for RCTs in MM between 2015 and 2020. Study design, characteristics of MM and its treatment, the primary outcomes, and the types of PRO instrument(s) were extracted using a predefined template. Additionally, in a stepwise approach, it was assessed whether the identified instruments had been validated for multiple myeloma patients, patients with haematological malignancies, or cancer patients. Results: Following screening for RCTs, 283 studies were included for review from 10,707 records retrieved, and 118 of these planned the use of PRO measures. Thirty-eight PRO instruments were reported. The most frequently used instrument (92 studies) was the EORTC QLQ-30. The EORTC-MY20 MM-specific questionnaire was the second most frequently used (50 studies), together with the EQ-5D (50 studies). Only 19 PRO instruments reported were consistent with the trial registry. Furthermore, in 58 publications, the information on PRO instruments differed between the publication and the trial registry. Further, information on PRO in HTA reports was available for 26 studies, of which 18 reports were consistent with the trial registries. Out of the 38 instruments used, six had been validated for patients with multiple myeloma (the most frequently used), six for patients with haematological malignancies, and 10 for cancer patients in general. Conclusions: The findings indicate that the measurement of PROs in RCTs for MM is underutilised, underreported, and often inconsistent. Guidelines for the appropriate use of PROs in MM are needed to ensure standardisation in selection and reporting. Furthermore, not all PRO instruments identified have been validated for myeloma patients or patients with haematological malignancies. Thus, guidelines for the appropriate use and reporting of PROs are needed in MM to ensure standardisation in the selection and reporting of PROs.

## 1. Introduction

### 1.1. Multiple Myeloma: The Disease

Multiple myeloma (MM) is a haematological malignancy of plasma cells with bone marrow and extra-medullary involvement, prevalently bone [1]. The incidence rate of MM is increasing worldwide, with an estimated incidence in Europe of 4.5–6.0/100,000/y. It comprises 1–1.8% of all cancers and is the second most frequent haematological malignancy [2]. Among patients with MM, approximately 73% have anaemia, 79% have an osteolytic bone disease with hypercalcemia, and 19% have acute kidney injury at the time of presentation [3]. Severe infections may occur due to secondary impaired immune function [4]. MM is most frequently diagnosed among people aged 65 to 74 years, with the median age being 69 years [5]. Despite the meaningful improvement in patients’ survival over the past decade, it is an incurable disease [2].

### 1.2. Treatment for Multiple Myeloma

Historically, the median overall survival (OS) for MM patients was only ~3 years, and there were a limited number of treatments available [6]. During the last two decades, tremendous advances have been made in the understanding of myeloma biology and the development of improved treatment strategies [2,7]. There is an increasing number of novel therapeutic agents and combinations that are contributing to improving patient outcomes, with substantial improvements observed mostly in patients up to 59 years of age [8]. Notably, increasing age correlates with poorer survival rates [9]. Thus, in a non-curative setting, the achievement of an improvement in survival and in quality of life (QoL) represent the primary goals of MM treatment [10,11].

### 1.3. Health-Related Quality of Life

Patients with MM report high symptom burden and reduced QoL. Bone lesions cause pain, fatigue, reduced role functioning, and body shape changes. Furthermore, bone marrow, renal failure, and immunodeficiency contribute to subsequent complications that impact QoL. These aspects, together with the psychosocial burden of a cancer diagnosis, may have varying importance for patients during different disease states [12,13,14]. The impact of MM on symptom burden and on QoL is worse compared to that of patients with other haematological malignancies [15,16].

In addition, the treatment of MM is toxic in itself and has a negative impact on patients’ QoL. It may be accompanied by multiple side effects [17,18]. The impact of new treatments on quality of life (QoL) is now an important component of decision-making in relapsed MM [19]. Thus, the optimisation of QoL of patients with MM is an important treatment goal, and the incorporation of PROs into clinical and research practice has the potential to improve treatment outcomes [20,21]. The assessment of patient-reported outcomes (PROs), such as QoL, symptoms, and treatment satisfaction, could provide additional information to more robustly inform patient care.

Patient-reported outcomes (PROs) are becoming increasingly important in clinical trials to highlight the intended benefits according to patients’ perspectives. In particular, in MM, PRO measurement is useful to reveal how treatment affects physical, psychosocial, and functional behaviour as well as symptoms and treatment-related adverse events to evaluate the benefit-risk ratio of a particular drug or drug combination.

Noteworthy, regulatory agencies for drug approval, such as US Food and Drug Administration (FDA) and European Medicines Agency (EMA), recommend PROs as a prioritised treatment outcome [22]. The relevance of PRO assessments in clinical trials of patients with hematologic malignancies has been previously reported [18,23,24]. Patient-reported outcome data may provide additional information about the benefits/risks of drugs from the patients’ perspective, which could not have been otherwise inferred by clinician-reported symptomatic adverse events. In a systematic review of QoL data from MM clinical trials, it was demonstrated that moderate to large QoL improvements can be expected with effective first-line treatments [16]. In addition, QoL improvement is included in the European Society of Medical Oncology Magnitude of Clinical Benefit Scale (ESMO-MCBS) to determine the “value” of a new therapeutic strategy in cancer [25].

Several systematic reviews (SR) on the use of PROs in MM clinical trials have been published [26,27,28]. For MM trials, it is unclear which PRO instruments are utilised, how often they are used, and whether results are consistently reported. A recent systematic review (SR) of registrational trials from 2007 to 2020 shows that PRO measurement methodology in MM differs significantly between trials which might negatively affect the comparability of PROs for different treatment regimes. Furthermore, only 68% of the trials contained submitted PRO data. Nine PRO measures were used, most commonly the EORTC QLQ-30, EQ-5D, and QLQ-MY20 [27]. In a recent review by Efficace and colleagues, it was stated that more consistency in the methodological approach to PRO assessment and interpretation of outcomes is needed to ensure that PRO findings will be more impactful on patient care [26].

The development of guidelines for the use of PROs in adult patients with haematological malignancies was supported and conceptualised by the European Hematology Association (EHA). The first step is the reporting of this systematic review that explores the use, frequency, and consistency of reporting of PRO instruments in randomised controlled trials (RCTs) for MM.

## 2. Materials and Methods

### 2.1. Data Sources

This study followed the 2020 PRISMA updated guideline for reporting systematic reviews [29]. This methodological review is not registered at PROSPERO, as the stepwise approach to identify PRO instruments used in trials evaluating patients with multiple myeloma and then to analyse whether the instruments used have been validated for patients with multiple myeloma, haematological malignancies of patients with cancer does not fit to the PROSPERO registration criteria. A search specialist systematically searched the databases Medline (Ovid) and CENTRAL for the period from 1 January 2015 to April 2021.

In the first step of this project, we aimed to identify scales and instruments that were most frequently used in MM trials to measure PROs. We systematically searched MEDLINE and CENTRAL for randomised controlled trials investigating MM between 2015 and 2020. Next, in order to find out which of the identified instruments were validated in MM patients, we conducted three separate literature searches. The rationale for this strategy was to report on psychometric validity for questionnaires frequently used for MM patients but not validated for this population. First, we identified which questionnaires have been validated for patients with MM. Second, for questionnaires not validated in MM patients, we conducted a search for studies validating the questionnaire for patients with haematological malignancies (HM). Finally, for questionnaires not validated in either MM or HM populations, we conducted a search for validation studies for all cancer types. Additionally, we performed hand searches for the instrument manuals and for any information not available in the identified publications.

The types of articles that were included were limited to RCTs that were published in peer-reviewed journals in the English language. Articles excluded were ‘grey’ literature, including dissertations, reports, editorials, letters to editors, pre-prints, commentaries, reviews, conference abstracts, and conference proceedings. Thus, the search was comprehensive, systematic, and well-documented. EndNote20^®^ was used to keep track of references following the published guidance [30]. Rejected studies were recorded with reasoning.

### 2.2. Search Strategy/Selection/Exclusion Criteria

The sensitive search strategy to identify randomised controlled trials evaluating multiple myeloma patients was developed by an information specialist and is added in Appendix A. MEDLINE and Cochrane Register of Controlled Studies were searched.

Types of studies included were RCTs (including cross-over trials and trials with open-label extensions if initial treatment was continued after study completion and those registered on clinicaltrial.gov) and used PROs as a primary or a secondary outcome). Study design, disease and treatment characteristics, the primary outcome, and PRO instrument(s) that were utilised were extracted using a pre-defined template.

Inclusion criteria included any adult, gender, ethnicity, setting, or country, while studies that recruited adolescents (under 18 years of age) were excluded. Interventions included any drug, therapeutic intervention, and alternative medicines, e.g., acupuncture, fire needle, Chinese traditional (herbal) medicine, Ayurvedic, and educational or lifestyle interventions, and the condition included was multiple myeloma.

After the identification of instruments used, a stepwise search approach was developed to assess whether the instrument had been validated for multiple myeloma patients (for search strategy, see Appendix B) or, if not, for patients with haematological malignancies (see Appendix C) or, if not, for patients with cancer (see Appendix D for search strategy)

Information on populations with which the instrument was validated, instrument characteristics, validity, reliability, sensitivity, and Minimally Clinically Important Difference (MCID) were extracted using a pre-defined template. If no validation for the original language version of a PRO instrument was found for MM, HM, or cancer, we resorted to validation studies in other languages. Search counts for inclusions and exclusions and reasons for the exclusion of studies were recorded in a PRISMA flowchart [29]. Each search was screened by two independent researchers (NK, MA) who discussed any disagreement at the end of the screening process and reached a consensus by discussion. Any unresolved matter was mediated by an adjudicator (NS).

### 2.3. Outcome Measures Extracted

The evaluation of PROs as an outcome measure (primary, secondary, or exploratory) was assessed.

The line of treatment for MM was evaluated and reported. For clinical trials, the outcome measure of the association of PRO change and response to treatment and whether subgroup analysis according to response was evaluated and whether the association between the respective PRO and response was significant.

### 2.4. Data Extraction and Synthesis

For data extraction, the guidance of the Cochrane Handbook for Systematic Reviews of Interventions was followed [31]. An Excel extraction template was created based on the main outcomes of the review. Checklist of the Cochrane Handbook Version 6.2 of items to consider in data collection or data extraction [32]. MA and NK independently extracted data from the included publications to parallel Excel database tables, and an adjudicator (NS) resolved any disagreements in the data extraction. Missing data were noted in the data templates, but none was deemed sufficiently important to contact authors for unreported data or additional details. To assess the consistency of PRO reporting, study registries were compared with publications and study protocol, and Health Technology Assessment (HTA) reports where available.

## 3. Results

Overall, 10,707 records were identified (see Prisma diagram, Figure 1). Following screening for RCTs, 283 ongoing, completed, or published studies were included for review. A total of 118 studies planned to assess PROs, and of these, 95 studies fully matched the search criteria. Sixty-three were still ongoing and had not reported on PRO outcomes according to therapeutic response.

Overall, 38 different PRO instruments were reported (Appendix E). The most frequently used PRO instrument (92 studies) was the EORTC QLQ-30 (Table 1, Figure 2) which measures QoL in cancer patients [33], and it has been validated in multiple myeloma [34]. The EORTC-MY20, a disease-specific tool for patients with MM, was also frequently used (50 studies) [35]. Similarly, the EQ-5D was used in 50 studies, and 35 other different PRO instruments were used in the remaining studies (Table 1). Validation studies in MM have been reported only for six instruments, including EORTC QLQ-30 and EORTC-MY20, the Myeloma Patient Outcome Scale (MyPOS), EORTC-MY24, MDASI MM, and FACT-MM.

Of 283 studies, 40% included subjects on first-line treatment, 42% on 2nd line treatment, and others were evaluating both or unknown. For less than half of the studies, a protocol was found. In 39 studies for which a study protocol was found, only 19 reported PRO instruments consistently with the trial registry for the study. In addition, HTA reports of 26 studies included a PRO, of which 18 reports were consistent with the trial registries. Thus, the PRO assessment was consistent with the protocol by only 20%. Of those trials for which the protocol defined a PRO as an outcome, more than half did not publish the results of the PRO measures (Figure 3).

Sixteen trials reported PRO changes but did not associate changes according to response, and the remaining 16 reported changes according to subgroups of response. The PRO measures correlated with therapeutic response in only one study [36].

Overall, 6 out of 38 instruments assessed in trials with MM patients between 2015 and 2020 are currently validated for MM patients. Likewise, six instruments are currently validated for HM patients. This includes the EQ-5D, one of the three instruments most frequently assessed in trials with MM patients. Finally, nine studies were validated in different cancer types (e.g., breast cancer, prostate cancer, or not further classified). Instruments for which no specific validation for patients with multiple myeloma, haematological malignancies, or cancer could be found are presented in Appendix F.

## 4. Discussion

Increasingly, PRO is considered an important endpoint in oncological trials that may improve the knowledge and care of cancer patients [37]. However, the number of haematological malignancy clinical trials reporting PROs is far less compared to RCTs involving other cancer patients. Patient-focused care is key to the optimal management of patients with MM, and it is becoming increasingly important to consider QoL and other PROs in such patients. In MM, obtaining the most durable remissions with the best QoL is the primary goal of MM treatment. Therefore, PRO measures are important endpoints in clinical trials aimed at assessing the efficacy of MM treatment. Quality of life data also plays an important role in health technology and cost-effectiveness assessments of new treatments in patients with MM. According to recent reviews of MM-RCTs, assessing PROs among other endpoints, the methodology of their collecting and reporting must be improved if the results might actually influence clinical decision-making [27,28,38]. In order to overcome these drawbacks, guidelines for the use of PROs in RCTs in haemato-oncology are worthwhile. For this purpose, the EHA initiated a project on the development of evidence-based guidelines for the use of PROs in adult patients with haematological malignancies. In this SR, we explored the use and frequency of PROs in RCTs for MM, identified the tools validated for MM, and assessed the consistency of their reporting.

Based on systematically searched databases such as Medline (Ovid), and CENRAL covering the period from 1 January 2015 to 30 November 2020, 95 RCTs using PROs for MM were identified. The most frequently used PRO tool was the EORTC QLQ-30 (92 studies). Other common PROs were the EORTC-MY20 and the EQ-5D (both were used in 50 studies). Altogether 38 PROs were identified. Only six PROs used in MM-RCTs were validated for MM. In the previous reviews, the EORTC QLQ-30, QLQ-MY20, and EQ-5D were used (65%), and (47%), respectively. In a recent review by Fernandes and colleagues, nine unique PRO measures were used, most commonly the EORTC QLQ-30 (87%), EQ-5D (65%), and QLQ-MY20 (47%) [27].

As for the consistency of reporting of PRO data, for less than half of the studies, a protocol was found. In 39 studies for which a study protocol was found, only 19 reported PRO instruments consistently with the trial registry for the study. In other words, the PRO assessment was consistent with the protocol by only 20%. For those trials for which the protocol defined a PRO as an outcome, more than half did not publish the results of the PRO measures. It is worth mentioning that the identified RCTs were heterogeneous regarding the presentation, analysis, and interpretation of PRO results. This finding supports the outcomes of a recent review by Efficace and colleagues [26]. Substantial heterogeneity in terms of PRO collection methods, measures, analyses, and interpretation may hinder the ability to effectively capture and interpret patient experience in future MM clinical trials.

## 5. Conclusions

The findings indicate that the measurement of PROs in RCTs for MM is underutilised and underreported. Furthermore, the reporting of PROs in the RCTs is often inconsistent. According to our findings, guidelines for the appropriate use of PROs in MM are needed to ensure standardisation in selection and reporting.

## 6. Future Considerations

In the era of precision/personalised medicine, where prototype technology is being applied to patient care, there is an even greater need for the assessment of patient-reported outcomes to become the standard of patient care. The current landscape of personalised medicine has the potential of creating a fundamental paradox that the majority would not be comfortable with being exposed to experimental diagnostic tests and the payers accepting the reality of pouring more money into a smaller group of patients. Thus, it is incumbent on us to move towards the development of universal, robust techniques with high precision for the assessment and reporting of patient-reported outcomes that lend themselves to clinical trials of personalised medicines.

## Figures and Tables

**Figure 1 cancers-14-06007-f001:**
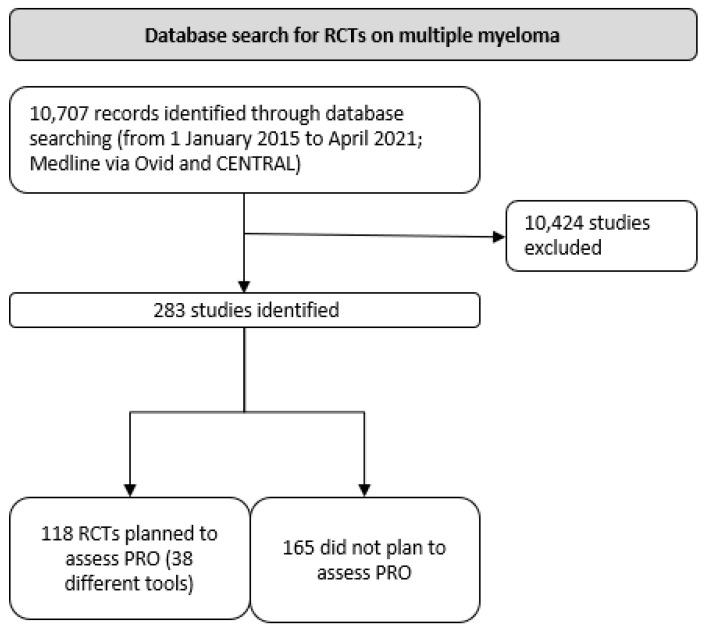
PRISMA flow diagram.

**Figure 2 cancers-14-06007-f002:**
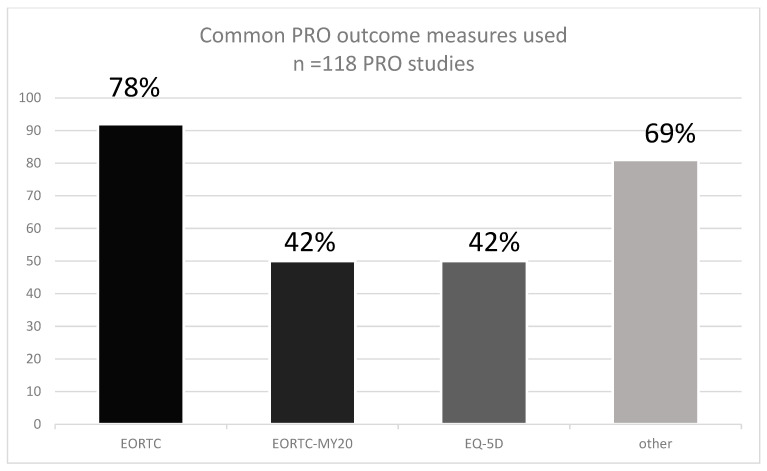
The prevalence of PRO instruments used in MM RCTs according to the SLR.

**Figure 3 cancers-14-06007-f003:**
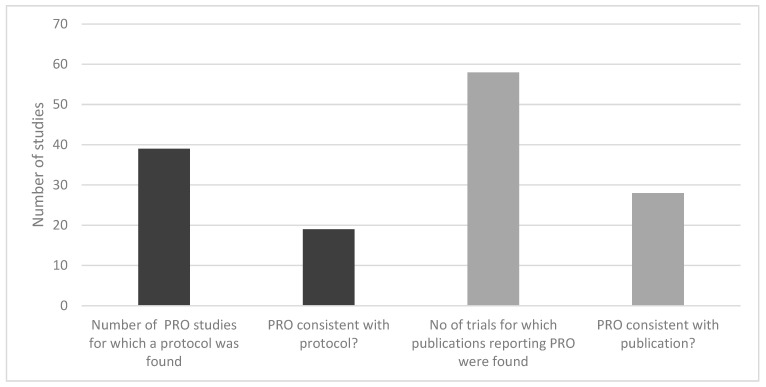
Consistency of reporting PRO instruments in MM RCTs.

**Table 1 cancers-14-06007-t001:** Overview of PRO instruments and validated populations.

Instrument	Frequency of (Planned) Assessment in MM Trials (N = 283)	Validated for Patients with Multiple Myeloma	Validated for Patients with Haematological Malignancies	Validated for Different Cancer Types
EORTC QLQ-MY20	50	+		
EORTC QLQ MY24	6			
EORTC QLQ C30	92	+		
MyPOS	1			
MDASI-MM	1			
FACT-MM	2	+		
EQ5D-3L	50			
EQ5D-5L				
HADS	2			
FACT-G	8			
FACT-BMT	5			
FACIT Fatigue Scale	7			
FACIT-Sp	1			
BPI-SF	7			
CTSQ	1			
FACT GOG-Ntx	7			
IPAQ-SF	1			
LANSS	1			
PSF-R	1			
PROMIS	2			
PRO-CTCAE	10			
SF-36	7			

Note. + indicate the availability of MCIDs for MM patients; the ligher the color the less relevant instrument.

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
