# Peer review of "The Reporting, Use, and Validity of Patient-Reported Outcomes in Multiple Myeloma in Clinical Trials: A Systematic Literature Review"

_cancers, 2022, doi:10.3390/cancers14236007_

Round 1

Reviewer 1 Report

This manuscript provide a systematic review of literature about PRO reporting in the context of multiple myeloma. It show that PRO is often not mentioned as a results in trial and also that adequate PRO may be not used by the investigatot. This finding are very important because they QaoL may be as important that efficiency for our patient .

I have several questions:

-        Systematic review included various therapeutic interventions, from new drug to alternative medicine. Did PRO report differ we consider different type of study or different geographic region?

-        Quality of publications may be heterogenous and have different impact. Is PRO reporting associated with higher impact factor?

-        What would be the recommendations of the author to a better implementation as PRO as a primary or secondary objective?

-        PRO should be simple to be sure that every patients answer the questionnaire. Would it explain the low rate of PRO reporting in the case of multiple myeloma? Do the authors recommend one particular PRO?

Author Response

  1. Comment:
  • Question:  Systematic review included various therapeutic interventions, from new drug to alternative medicine. Did PRO report differ we consider different type of study or different geographic region?
  • Response: We confirm that we considered all types of Randomized Clinical Trials from any geographic region, without restrictions.
  1. Comment:
  • Question: Quality of publications may be heterogenous and have different impact. Is PRO reporting associated with higher impact factor?
  • Response: PRO reporting in clinical trials is considered an important contribution to the effects of treatment from a patient perspective and is required for the approval of new drugs. Therefore, it is associated with a higher impact in terms of the added value for clinical decision making.
  1. Comment:
  • Question: What would be the recommendations of the author to a better implementation as PRO as a primary or secondary objective?
  • Response: We thank the reviewer for this request. In fact, the recommendations deriving from this work and from other sources are the subject of another publication. This has been suggested in our remark in our Conclusions on the need for Guidelines.
  1. Comment:
  • Question: PRO should be simple to be sure that every patients answer the questionnaire. Would it explain the low rate of PRO reporting in the case of multiple myeloma? Do the authors recommend one particular PRO?
  • Response: The low rate of PRO reporting is not only due to compliance of patients but may often be due to compliance of investigators. The recommendations of particular PROs in MM are the subject of another publication as suggested in our remark in our Conclusions on the need for Guidelines.

Reviewer 2 Report

It is a quite good review regarding a very important issue in Multiple Myeloma (MM) that is Patient Reported Outcomes (PRO) in Clinical Trials. Quality of life of these patients and the parameters have to be emphasized in the trials and the new treatments options. This present study and the literature review offers a global point of view for the existence of PRO in specifically MM patients.

According to the manuscipt I have reviewed I have to comment the following points:   Introduction with a well extended backround in MM patients and PRO.   Mehtothology quite good as a retrospective meta analysis regarding the studies for PRO in patients with MM. It highlights the current PRO measures in RCT for MM.   Five years is a good time period to study and review trials for MM.   In the results there is a clean and clarified attribution of the RCT and the validation in MM patients.   Discussion could be more extended for the review of the literature and the correct under utilized measurement of RPO in MM patients.   In conclusion it can be accepted for pointing out the lack of PRO in MM patients.

Author Response

According to the manuscript I have reviewed I have to comment the following points:   Introduction with a well extended background in MM patients and PRO.  Methodology quite good as a retrospective meta analysis regarding the studies for PRO in patients with MM. It highlights the current PRO measures in RCT for MM.   Five years is a good time period to study and review trials for MM.   In the results there is a clean and clarified attribution of the RCT and the validation in MM patients.   Discussion could be more extended for the review of the literature and the correct under utilized measurement of RPO in MM patients.   In conclusion it can be accepted for pointing out the lack of PRO in MM patients.

Response: We thank the reviewer for the encouraging comments and we have now extended the discussion, as suggested.